# Geo-PIFu: Geometry and Pixel Aligned Implicit Functions for Single-view Human Reconstruction

**Tong He**[1], **John Collomosse**[2,3], **Hailin Jin**[2], **Stefano Soatto**[1]

[1]UCLA. [2]Creative Intelligence Lab, Adobe Research. [3]CVSSP, University of Surrey, UK.

{simpleig,soatto}@cs.ucla.edu  {collomos,hljin}@adobe.com

## Abstract

We propose Geo-PIFu, a method to recover a 3D mesh from a monocular color image of a clothed person. Our method is based on a deep implicit function-based representation to learn latent voxel features using a structure-aware 3D U-Net, to constrain the model in two ways: first, to resolve feature ambiguities in query point encoding, second, to serve as a coarse human shape proxy to regularize the high-resolution mesh and encourage global shape regularity. We show that, by both encoding query points and constraining global shape using latent voxel features, the reconstruction we obtain for clothed human meshes exhibits less shape distortion and improved surface details compared to competing methods. We evaluate Geo-PIFu on a recent human mesh public dataset that is $10\times$ larger than the private commercial dataset used in PIFu and previous derivative work. On average, we exceed the state of the art by $42.7\%$ reduction in Chamfer and Point-to-Surface Distances, and $19.4\%$ reduction in normal estimation errors.

## 1 Introduction

Image based modeling is enabling new forms of immersive content production, particularly through realistic capture of human performance. Recently, deep implicit modeling techniques delivered a step change in 3D reconstruction of clothed human meshes from monocular images. These implicit methods train deep neural networks to estimate dense, continuous occupancy fields from which meshes may be reconstructed e.g. via Marching Cubes [20].

Reconstruction from a single view is an inherently under-constrained problem. The resulting ambiguities are resolved by introducing assumptions into the design of the implicit surface function; the learned function responsible for querying a 3D point's occupancy by leveraging feature evidence from the input image. Prior work extracts either a global image feature [21, 23, 3, 18], or pixel-aligned features (PIFu [29]) to drive this estimation. Neither approach takes into account fine-grain local shape patterns, nor seek to enforce global consistency to encourage physically plausible shapes and poses in the reconstructed mesh. This can lead to unnatural body shapes or poses, and loss of high-frequency surface details within the reconstructed mesh.

This paper contributes Geo-PIFu: an extension of pixel-aligned features to include three dimensional information estimated via a latent voxel representation that enriches the feature representation and regularizes the global shape of the estimated occupancy field. We augment the pixel-aligned features with *geometry-aligned shape features* extracted from a latent voxel feature representation obtained by lifting the input image features to 3D. These voxel features naturally align with the human mesh in 3D space and thus can be trilinearly interpolated to obtain a query point's encoding. The uniform coarse occupancy volume losses and the structure-aware 3D U-Nets architecture used to supervise and generate the latent voxel features, respectively, both help to inject global shape topology robustness / regularities into the voxel features. Essentially, the latent voxel features serve as a coarse human shape proxy to constrain the reconstruction. Figure 1 summarises our proposed adaptation (see upper

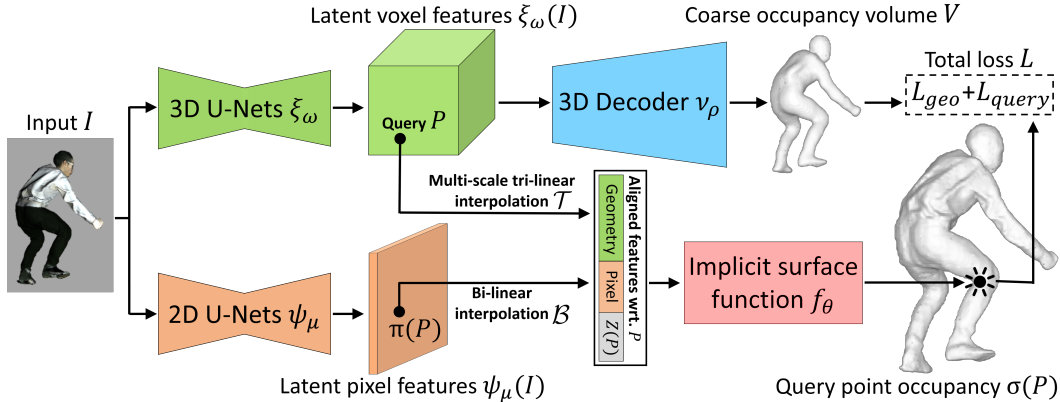

Figure 1: Pipeline. Our method extracts latent 3D voxel and 2D pixel features from a single-view color image. The extracted features then are used to compose geometry and pixel aligned features *wrt.* each query point $P$ for occupancy estimation through an implicit surface function. At training time, we enforce losses on both the coarse occupancy volume and the estimated query point occupancy values. Note that the blue color 3D convolution decoder for generating the coarse occupancy volume is only needed at training time to supervise the latent voxel features.

branch) to reconstruct a clothed human mesh from a single-view image input using a geometry and pixel aligned implicit surface function. Our three technical contributions are:

**1. Geometry and pixel aligned features.** We fuse geometry (3D, upper branch) and pixel (2D, lower branch) aligned features to resolve local feature ambiguity. For instance, query points lying upon similar camera rays but on the front or back side of an object are reprojected to similar pixel coordinates resulting in similar interpolated 2D features. Incorporating geometry-aligned shape features derived from our latent voxel feature representation resolves this ambiguity, leading to clothed human mesh reconstructions with rich surface details.

**2. Global shape proxy.** We leverage the same latent voxel feature representation as a coarse human shape proxy to regularise reconstruction and encourage plausible global human shapes and poses; crucial given the requirement to hallucinate the unobserved rear of the imaged object. This improves the plausibility of the estimated character shape and pose with accurate ground truth 3D alignment, and reduces mesh artifacts like distorted hands and feet.

**3. Scale of Evaluation.** We extend the evaluation of deep implicit surface methods to the DeepHuman dataset [38]: 5436 training meshes, 1359 test meshes — 10 times larger then the private commercial dataset used in PIFu. The leading results on this dataset are generated by voxel and parametric mesh representations based methods. No deep implicit function method has been benchmarked on it before.

We show that Geo-PIFu exceeds the state of the art deep implicit function method PIFu [29] by $42.7\%$ reduction in Chamfer and Point-to-Surface Distances, and $19.4\%$ reduction in normal estimation errors, corresponding to qualitative improvements in both local surface details and global mesh topology regularities.

## 2 Related Work

Our method is most directly related to the use of deep implicit function representations for single-view mesh reconstruction. For context, we will also briefly review on explicit shape representations.

### 2.1 Implicit Surface Function

Occupancy Networks [21], DeepSDF [23], LIF [3] and DIST [18] proposed to use global representations of a single-view image input to learn deep implicit surface functions for mesh reconstruction. They demonstrated state-of-the-art single-view mesh reconstruction results on the ShapeNet dataset [2] of rigid objects with mostly flat surfaces. However, the global representation based implicit function does not have dedicated query point encodings, and thus lacks modeling power for articulated parts and fine-scale surface details. This motivates later works of PIFu [29] and DISN [36]. They

utilize pixel-aligned 2D local features to encode each query point when estimating its occupancy value. The alignment is based on (weak) perspective projection from query points to the image plane, followed by bilinear image feature interpolation. PIFu for the first time demonstrated high-quality single-view mesh reconstruction for clothed human with rich surface details, such as clothes wrinkles. However, PIFu still suffers from the feature ambiguity problem and lacks global shape robustness. Another two PIFu variations are PIFuHD [30] and ARCH [10]. PIFuHD leverages higher resolution input then PIFu through patch-based feature extraction to accommodate GPU memory constraints. ARCH combines parametric human meshes (*e.g.* SMPL [19]) with implicit surface functions in order to assign skinning weights for the reconstructed mesh and enable animations. Both methods require more input / annotations (e.g. $2\times$ higher resolution color images, SMPL registrations) than PIFu. Note that PIFuHD, ARCH and Geo-PIFu focus on different aspects and are complementary. If provided with those additional data, our method can also integrate their techniques to further improve the performance. Another related work that also utilizes latent voxel features for implicit function learning is IF-Net [4], but its problem setting is different from ours. While IF-Net takes partial or noisy 3D voxels as input, (Geo)-PIFu(HD) and ARCH only utilize a single-view color image. Thus, IF-Net has access to "free" 3D shape cues of the human subject. But Geo-PIFu must achieve an ill-posed 2D to 3D learning problem. Meanwhile, Geo-PIFu needs to factorize out pixel domain nuisances (*e.g.* colors, lighting) in order to robustly recover the underlying dense and continuous occupancy fields.

## 2.2 Explicit Shape Models

Explicit shape models can be classified by the type of 3D shape representation that they use: voxel, point cloud or (parametric) mesh [33, 12, 17, 5, 37, 8, 6, 34, 28, 19, 13, 38, 1]. Here we only review some representative works on single-view clothed human body modeling and reconstruction. BodyNet [33] leverages intermediate tasks (*e.g.* segmentation, 2D/3D skeletons) and estimates low-resolution body voxel grids. A similar work to BodyNet is VRN [12], but it directly estimates occupancy voxels without solving any intermediate task. Voxel-based methods usually are constrained by low resolution and therefore struggle to recover shape details. Driven by data efficiency and topology modeling flexibility of point clouds, several methods [17, 5, 37] propose to estimate point coordinates from the input image. The drawback is that generating a mesh with fine-scale surface details by Poisson reconstruction [14] requires a huge number of point estimations. In order to directly generate meshes, Atlas-Net [6] uses deep networks to deform and stitch back multiple flat-shape 2D grids. However, the reconstructed meshes are not water-tight because of the stitching hole artifacts. Pixel2Mesh [34] generates water-tight meshes by progressively deforming a sphere template. But the empirical results show that mesh-based methods produce overly smooth surfaces with limited surface details due to shape flexibility constraints of the template. Parametric shape models like SMPL [19], BlendSCAPE [9], *etc* are also useful for directly generating human meshes. For example, [13, 26, 16, 15, 25] estimate or fit the shape and pose coefficients of a SMPL model from the input image. Note that these parametric human body models are usually designed to be "naked", ignoring clothes shapes. Recently, methods that leverage hybrid explicit models are also proposed to combine the benefits of different shape representations. DeepHuman [38] and Tex2Shape [1] combine SMPL with methods of voxel refinement and mesh deformation, respectively, for single-view clothed human mesh reconstruction. One issue of using parametric body shapes is non-trivial data annotation (*e.g.* registering SMPL models with ground-truth clothed human meshes), which is not needed in deep implicit surface function-based methods. Another problem is propagation error: wrong SMPL estimation can cause additional errors in its afterward steps of local surface refinement.

## 3 Method

Continuous occupancy fields map each 3D point inside it to a positive or negative occupancy value dependent on the point is inside the target mesh or not. We encode surface as the occupied / unoccupied space decision boundary of continuous occupancy fields and apply the Marching Cube algorithm [20] to recover human meshes. Occupancy values are denoted by $\sigma \in \mathbb{R}$. Points inside a mesh have $\sigma > 0$ and outside are $\sigma < 0$. Thus, the underlying surface can be determined at $\sigma := 0$. In this work the goal is to learn a deep implicit function $f(\cdot)$ that takes a single-view color image $I : D \subset \mathbb{R}^2 \to \mathbb{R}^3$ and coordinates of any query point $P \in \mathbb{R}^3$ as input, *and* outputs the occupancy value $\sigma$ of the query point: $f \mid (I, P) \mapsto \sigma$. The training data for learning the implicit surface function

consists of (image, query point occupancy) pairs of $\{I, \sigma(P)\}$. More specifically, we densely sample training query points on the mesh and generate additional samples by perturbing them with Gaussian noise. In the following sections, we introduce the formulation of our deep implicit function in Section 3.1 and explain our training losses in Section 3.2. Implementation details are provided in Section 3.3.

## 3.1 Geo-PIFu

As shown in Figure 1, we propose to encode each query point $P$ using fused geometry (3D, upper branch) and pixel (2D, lower branch) aligned features. Our method, Geo-PIFu, can be formulated as:

$$f_\theta(\mathcal{T}(\xi_\omega(I), P), \mathcal{B}(\psi_\mu(I), \pi(P)), Z(P))) := \sigma \qquad (1)$$

The implicit surface function is denoted by $f_\theta(\cdot)$ and implemented as a multi-layer perceptron with weights $\theta$. For each query point $P$, inputs to the implicit function for occupancy $\sigma(P)$ estimation include three parts: geometry-aligned 3D features $\mathcal{T}(\xi_\omega(I), P)$, pixel-aligned 2D features $\mathcal{B}(\psi_\mu(I), \pi(P))$, and the depth $Z(P) \in \mathbb{R}$ of $P$. They compose the aligned features *wrt.* $P$

### 3.1.1 Geometry-aligned Features

First we explain geometry-aligned features $\mathcal{T}(\xi_\omega(I), P)$ in Equation (1). Here $\xi_\omega(\cdot)$ are 3D U-Nets [11] that lift the input image $I$ into 3D voxels of latent features. The networks are parameterized by weights $\omega$. To extract geometry-aligned features from $\xi_\omega(I)$ *wrt.* a query point $P$, we conduct multi-scale trilinear interpolation $\mathcal{T}$ based on the xyz coordinates of $P$. This interpolation scheme is inspired by DeepVoxels [31, 7] and IF-Net [4]. It is important to notice that these voxel features naturally align with the human mesh in 3D space and thus can be trilinearly interpolated to obtain a query point's encoding. Specifically, we trilinearly interpolate and concatenate features from a point set $\Omega$ determined around $P$.

$$\Omega := \left\{ P + n_i \mid n_i \in d \cdot \left\{ (0,0,0)^T, (1,0,0)^T, (0,1,0)^T, (0,0,1)^T ... \right\} \right\} \qquad (2)$$

Here $n_i \in \mathbb{R}^3$ is a translation vector defined along the axes of a Cartesian coordinate with step length $d \in \mathbb{R}$. For convenience, we will use trilinear interpolation to indicate our multi-scale feature interpolation scheme $\mathcal{T}$ for the rest of the paper. Not like global image features and pixel-aligned features, as discussed in Section 1, geometry-aligned features provide dedicated representations for query points that directly capture their local shape patterns and tackle the feature ambiguity problem. Since computation cost of latent voxel features learning is usually a concerned issue, we emphasize that our latent voxel features are of low resolution (C-8, D-32, H-48, W-32), in total 393216. In comparison, the latent pixel features resolution is (C-256, H-128, W-128), in total 4194304. More discussion on the model parameter numbers is provided in the experiment section.

### 3.1.2 Pixel-aligned Features

Pixel-aligned features are denoted as $\mathcal{B}(\psi_\mu(I), \pi(P))$ in Equation (1), where $\pi$ represents (weak) perspective camera projection from the query point $P$ to the image plane of $I$. Then pixel-aligned features can be extracted from the latent pixel features $\psi_\mu(I)$ by bilinear interpolation $\mathcal{B}$ at the projected pixel coordinate $\pi(P)$. Image features mapping function $\psi_\mu(\cdot)$ is implemented as 2D U-Nets with weights $\mu$. While geometry-aligned features, by designs of network architectures and losses, are mainly informed of coarse human shapes, pixel-aligned features focus on extracting high-frequency shape information from the single-view input, such as clothes wrinkles. In Section 4.2 we conduct ablation studies on single-view clothed human mesh reconstruction using individual type of features. The results empirically demonstrated these claimed attributes of geometry and pixel aligned features.

## 3.2 Training Losses

Next we explain the losses used in training the geometry and pixel aligned features, as well as the deep implicit surface function.

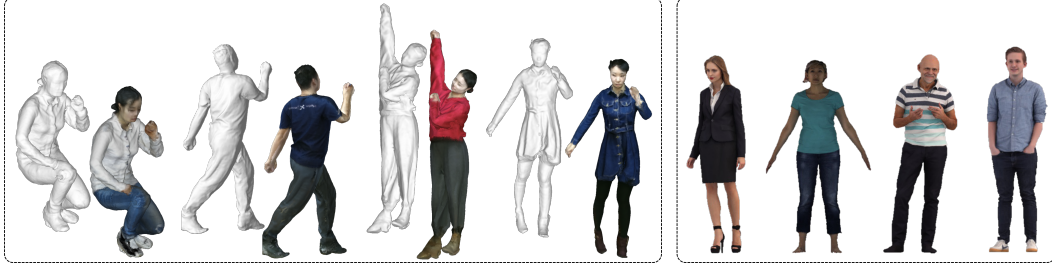

Figure 2: Left: the DeepHuman dataset contains various clothes and poses. Right: the dataset used in PIFu consists of mostly upstanding poses.

### 3.2.1 Coarse occupancy volume loss

To learn latent voxel features $\xi_\omega(I) \in \mathbb{R}^{m \times s_1 \times s_2 \times s_3}$ that are informed of coarse human shapes, we use a 3D convolution decoder and a sigmoid function to estimate a low-resolution occupancy volume $\hat{V}(I) \in \mathbb{R}^{1 \times d \times h \times w}$. We use extended cross-entropy losses [11] for training.

$$L_{geo} = -\frac{1}{|\hat{V}|} \sum_{i,j,k} \gamma V^{i,j,k} \log \hat{V}^{i,j,k} + (1-\gamma)(1-V^{i,j,k}) \log(1-\hat{V}^{i,j,k}), \text{ with } \hat{V}(I) = \nu_\rho(\xi_\omega(I)) \quad (3)$$

The ground truth coarse occupancy fields $V \in \mathbb{R}^{1 \times d \times h \times w}$ are voxelized from the corresponding high-resolution clothed human mesh. $(i, j.k)$ are indices for the (depth, height, width) axes, and $\gamma$ is a weight for balancing losses of grids inside / outside the mesh. $\nu_\rho(\cdot)$ represents the 3D convolution decoder and the sigmoid function used for decoding the latent voxel features $\xi_\omega(I)$. As shown in Figure 1, this 3D convolution decoder is only needed at training time to provide supervision for the latent voxel features. At inference time, we just maintain the latent voxel features for trilinearly interpolating geometry-aligned 3D features. Compared with learning dense continuous occupancy fields, this coarse occupancy volume estimation task is easier to achieve. The learned voxel features are robust at estimating shapes and poses for the visible side and also hallucinating plausible shapes and poses for the self-occluded invisible side. This is critical when the human has complex poses or large self-occlusion. Meanwhile, using structure-aware 3D U-Nets to generate these voxel-shape features also helps inject shape regularities into the learned latent voxel features.

### 3.2.2 High-resolution query point loss

The deep implicit surface function is learned by query point sampling based sparse training. It is difficult to directly estimate the full spectrum of dense continuous occupancy fields, and therefore, at each training step, we use a group of sparsely sampled query points (*e.g.* 5000 in PIFu and Geo-PIFu) to compute occupancy losses. As shown in Equation (1) as well as Figure 1, for each query point $P$ the deep implicit function utilizes geometry and pixel aligned features to estimate its occupancy value $\hat{\sigma}$. We use mean square losses in training.

$$L_{query} = \frac{1}{Q} \sum_{q=1}^{Q} (\sigma_q - \hat{\sigma}_q)^2 \quad (4)$$

The number of sampled query points at each training step is $Q$ and the index is $q$. The ground-truth occupancy value of a query point $P$ is $\sigma$. The total losses used in Geo-PIFu are $L = L_{geo} + L_{query}$. Following [22, 27] we adopt a staged training scheme of the coarse occupancy volume loss and the high-resolution query point loss. While $L_{geo}$ is designed to solve a discretized coarse human shape estimation task, $L_{query}$ utilizes sparse query samples with continuous xyz coordinates and learns high-resolution continuous occupancy fields of clothed human meshes. Experiments in Section 4.2 demonstrated that the query point loss is critical for learning high-frequency shape information from the input image, and enables the deep implicit surface function to reconstruct sharp surface details.

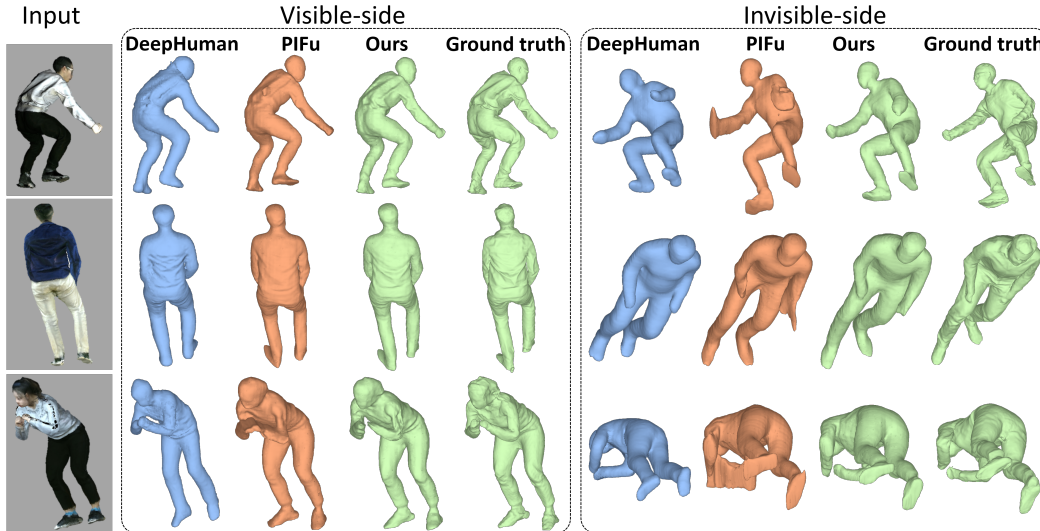

Figure 3: Single-view clothed human mesh reconstruction. Our results have less shape artifacts and distortions than PIFu. Besides improved global regularities and better alignment with ground truth, our meshes also contain more accurate and clear local surface details than DeepHuman and PIFu. DeepHuman can generate physically plausible topology benefiting from the voxelized SMPL model, but is incapable of capturing rich surface details due to limited voxel resolutions. Meanwhile, PIFu lacks global robustness when reconstructing meshes of complex poses and large self-occlusion.

## 3.3 Implementation Details

We use PyTorch [24] with RMSprop optimizer [32] and learning rate $1e - 3$ for both losses. We stop the 3D decoder training after 30 epochs and the implicit function training after 45 epochs using AWS V100-6GPU. The learning rate is reduced by a factor of 10 at the $8th$, $23th$ and $40th$ epochs. We use a batch of 30 and 36 single-view images, respectively. For each single-view image, we sample $Q = 5000$ query points to compute $L_{query}$. The step length $d$ for multi-scale tri-linear interpolation is 0.0722 and we stop collecting features at $2\times$ length. The balancing weight $\gamma$ of $L_{geo}$ is 0.7.

## 4 Experiments

We compare Geo-PIFu against other competing methods on single-view clothed human mesh reconstruction in Section 4.1. To understand how do the learned 3D-geometry and 2D-pixel aligned features impact human mesh reconstruction, we also show ablation studies on individual type of features and different feature fusion architectures in Section 4.2. Failure cases and effect of 3D generative adversarial network (3D-GAN) training are discussed in Section 4.3. More qualitative results and network architecture details can be found in supplementary.

**Dataset** We use the DeepHuman dataset [38]. It contains 5436 training and 1359 test human meshes of various clothes and poses. In comparison, the dataset used by PIFu [29] only has 442 training and 54 test meshes of mostly upstanding poses. Mesh examples are shown in Figure. 2. More importantly, our dataset is public and the meshes are reconstructed from cheap RGB-D sensors, while the PIFu dataset is private commercial data well-polished by artists. This means that our results can all be reproduced and the mesh collection procedures can be easily expanded to other domain-specific scenarios to obtain more human meshes (*e.g.* basketball players). Another critical difference is that we support free camera rotation when rendering the training and test images, as shown in the input column of Figure 3. In contrast, PIFu images are all rendered with zero elevation and in-plane rotation, namely the camera is always facing upright towards the mesh. In brief, our dataset is public and has more diversities in human subjects, poses, clothes and rendering angles. Therefore, our dataset is more challenging to learn and less likely to cause over-fitting on upstanding human poses and horizontal camera angles than the dataset used in PIFu. The downside of the DeepHuman dataset is that it lacks high quality texture maps for photo-realistic rendering, which might hurt model generalization on in-the-wild natural images.

Table 1: Benchmarks. These methods are all trained with the same DeepHuman dataset for fair comparisons. To evaluate global topology accuracy of meshes, we report CD ($\times 10000$) and PSD ($\times 10000$) between the reconstructed human mesh and the ground truth mesh. We also compute Cosine and *L2* distances for the input view normals to measure fine-scale surface details, such as clothes wrinkles. Small values indicate good performance. Our approach outperforms the competing methods, demonstrating the global / local advantages of utilizing geometry and pixel aligned features for deep implicit surface function learning.

| | Mesh | | Normal | |
|---|---|---|---|---|
| | CD | PSD | Cosine | *L2* |
| DeepHuman | 11.928 | 11.246 | 0.2088 | 0.4647 |
| PIFu | 2.604 | 4.026 | 0.0914 | 0.3009 |
| Ours | **1.742** | **1.922** | **0.0682** | **0.2603** |

Table 2: Benchmarks. These two models are evaluated using pre-trained weights provided by their authors. Parameter size of Geo-PIFu is 30616954 (*12 times smaller than PIFuHD*).

| Method | Parameter Size | Mesh | | Normal | |
|---|---|---|---|---|---|
| | | CD | PSD | Cosine | *L2* |
| PIFu | 15604738 | 10.571 | 9.285 | 0.1422 | 0.4141 |
| PIFuHD | 387049625 | 9.489 | 9.349 | 0.1228 | 0.3776 |

**Metrics** To evaluate reconstructed human meshes, we compute Chamfer Distance (CD) and Point to Surface Distance (PSD) between the reconstructed mesh and the ground truth mesh. While CD and PSD focus more on measuring the global quality of the mesh topology, we also compute Cosine and *L2* distances upon the mesh normals to evaluate local surface details, such as clothes wrinkles. These metrics are also adopted in PIFu for human mesh reconstruction evaluation.

**Competing Methods** Recently DeepHuman [38], PIFu [29] and PIFuHD [30] have demonstrated SOTA results on single-view clothed human mesh reconstruction.

**1) DeepHuman** integrates the benefits of parametric mesh models and voxels. It first estimates and voxelizes a SMPL model [19] from a color image, and then refines the voxels through 3D U-Nets. To improve surface reconstruction details, they further estimate a normal map from the projected voxels and use the estimated normals to update the human mesh. DeepHuman shows large improvement over both parametric shapes and voxel representations. Therefore we compare Geo-PIFu against it. Note that this is not a fair comparison, because DeepHuman requires registered SMPL parametric meshes as additional data annotation.

**2) PIFu** leverages pixel-aligned image features to encode each query point for implicit surface function-based continuous occupancy fields learning. This method is most related to our work since we both use implicit surface function for 3D shape representation. Note that PIFu and Geo-PIFu have the same requirement of color image inputs and ground-truth training meshes.

**3) PIFuHD** is built upon PIFu with higher resolution inputs than all other competing methods and than our method. It also uses offline estimated front/back normal maps to further augment input color images. These additional modules make PIFuHD a parameter-heavy model (see Table 2).

### 4.1 Single-view Reconstruction Comparisons with State-of-the-art

**Global Topology** Results of CD / PSD between the recovered human mesh and the ground truth mesh are provided in the left column of Table 1. As explained in the metrics sections, these two metrics focus more on measuring the global quality of the mesh topology. Our results surpass the second best method PIFu by $42.7\%$ relative error reduction on average. Such improvement indicates that human meshes recovered by our method have better global robustness / regularities and align better with the ground truth mesh. These quantitative findings as well as analysis have also been visually proved in Figure 3. Our mesh reconstructions have fewer topology artifacts and distortions (*e.g.* arms, feet) than PIFu, and align better with the ground truth mesh than DeepHuman. Moreover, we include results of pre-trained models released by PIFu and PIFuHD in Table 2, because the latter has not yet released its training scripts. Under the same training data, PIFuHD achieves lower relative improvement over PIFu than Geo-PIFu. Note that the two ideas of PIFuHD (using sliding windows to ingest high resolution images, and offline estimated front/back normal maps to further augment

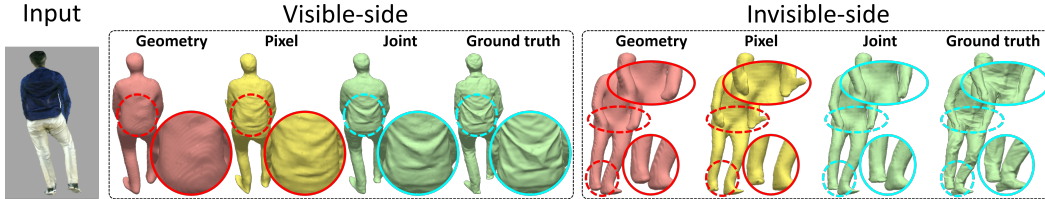

Figure 4: Left to right: mesh reconstruction results of exp-a, b, e in Table 3. This comparison shows that mesh reconstructions using only the 3D geometry features have better global topology regularities and ground truth alignment, but contain fewer local surface details than meshes reconstructed from only the 2D pixel features. The best global / local performances are achieved when two types of representations are jointly used for deep implicit surface function learning.

Table 3: Ablation studies. To analyze impact of the learned latent voxel, pixel features, we report human mesh reconstruction results that use individual type of features and the fused features. Small values indicate good performance. Exp-a, b evaluate meshes reconstructed from solely 3D geometry or 2D pixel features, respectively. Exp-c, d, e are results based on different feature fusion architectures for the geometry and pixel aligned features. Note that exp-b is the same as PIFu in Table 1, just named differently. Because essentially PIFu is a degenerate case of our method which only extracts pixel-aligned representations when learning implicit surface functions. Our results in the benchmarks are generated by the exp-e configuration. More analysis is in Section 4.2.

|  | Mesh | | Normal | |
|---|---|---|---|---|
|  | CD | PSD | Cosine | L2 |
| a. Ours-geometry | 2.293 | 2.629 | 0.0864 | 0.3164 |
| b. Ours-pixel | 2.604 | 4.026 | 0.0914 | 0.3009 |
| c. Ours-late-3D | 1.753 | 1.930 | **0.0675** | **0.2598** |
| d. Ours-late-both | **1.677** | **1.868** | 0.0764 | 0.2767 |
| e. Ours-early-3D | **1.742** | **1.922** | **0.0682** | **0.2603** |

input color images) are both add-on modules *wrt.* (Geo)-PIFu. Given high resolution images and offline estimated normal maps, one might combine PIFuHD with Geo-PIFu for further improved local surface details and global topology robustness. But this is out of the scope of our work.

**Local Details** Besides improved global topology, our mesh reconstruction results contain more local surface details such as clothes wrinkles. The ability of capturing fine-scale surface details is quantitatively measured in the right column of Table 1. We compute Cosine and $L2$ distances between the input view normals of the reconstructed mesh and those of the ground truth mesh. Our approach improves over the second best method PIFu by $19.4\%$ relative error reduction on average. To further explain the improved global and local performances of our method, we conduct ablation studies on individual type of features and different feature fusion architectures in the next section.

## 4.2 Ablation Studies

**Geometry vs. Pixel Aligned Features** In our method, the implicit function utilizes both the geometry-aligned 3D features and the pixel-aligned 2D features to compose query point encodings for occupancy estimation. To understand the advantages of our method, we separately show the impact of these two different types of features on clothed human mesh reconstruction. Results are reported in exp-a, b of Table 3. Ours-geometry and ours-pixel reconstruct human meshes using solely the 3D geometry or 2D pixel features, respectively. Exp-a shows significant improvement over exp-b in PSD, indicating that meshes reconstructed from 3D geometry features have better global topology robustness / regularities. This is attributed to global structure-aware 3D U-Net architectures and uniform coarse occupancy volume losses used to generate and supervise the latent voxel features. Visual comparisons in Figure 4 also verified this argument. However, meshes of exp-a are overly smooth lacking local surface details. This means that 2D pixel features focus more on learning high-frequency shape information such as clothes wrinkles. Moreover, by comparing exp-a, b with exp-e in Table 3 and Figure 4, we observe that the best global and local performances are achieved only when 3D geometry and 2D pixel features are fused. Namely, the fused features contain the richest shape information *wrt.* each query point for implicit surface function-based dense, continuous occupancy fields learning.

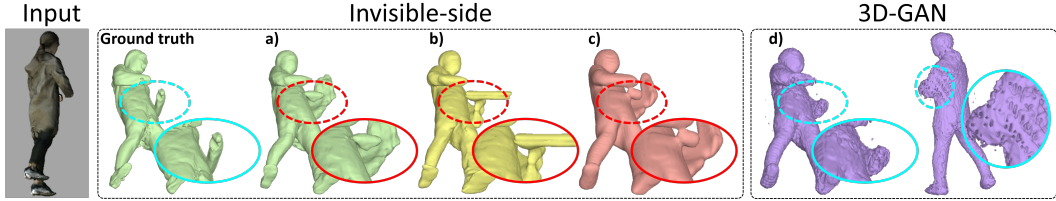

Figure 5: Typical failure cases. Plot-a is our method utilizing the early fusion based geometry and pixel aligned features. Plot-b, c are meshes reconstructed from solely the latent pixel or voxel features, respectively. Plot-d is similar to plot-c, but uses latent voxel representations trained jointly with coarse occupancy volume losses and 3D-GAN losses. When the latent pixel and voxel features, separately, both lead to human meshes with severe artifacts, the fused features struggle to correct all of these shape distortions. Further analysis on plot-d is presented in Section 4.3.

**Feature Fusion Architectures** Knowing that the integrated representations lead to the best mesh reconstruction performance, we now further explore different 3D / 2D feature fusion architectures. Results are reported in exp-c, d, e of Table 3. Exp-c Ours-late-3D applies several fully connected (FC) layers upon the geometry-aligned 3D features before concatenating them with the pixel-aligned 2D features. Exp-d Ours-late-both applies separate FC layers on both the geometry and pixel aligned features before concatenating them. Exp-e Ours-early-3D directly concatenate the 3D / 2D features, as shown in Figure 1. Comparisons on CD, PSD, Cosine and $L2$ normal distances mean that the mesh reconstruction results are not quite sensitive to the fusion architectures. All these different feature fusion methods demonstrate clear improvement over exp-a, b which only use a single type of features for human mesh reconstruction. In our main benchmark results we use the configuration of exp-e for its balanced mesh and normal reconstruction performances and its simple implementation.

## 4.3 Failure Cases and Adversarial Training

Typical failure cases are shown in Figure 5. Plot-a is our method. Plot-b, c are meshes reconstructed from solely the 2D pixel features or the 3D geometry features, respectively. We have demonstrated in Figure 4 and Table 3 that mesh topology artifacts and wrong poses can be corrected leveraging global shape regularities from the geometry-aligned 3D features. However, in cases where 2D, 3D features (*i.e.* plot-b, c) both fail to generate correct meshes, the fused features (*i.e.* plot-a) will still lead to failures. To address this problem, we turn to 3D generative adversarial networks (3D-GAN). The implicit surface function training is based on sparse query points sampling and thus non-trivial to enforce 3D-GAN supervision directly on the whole continuous occupancy fields. Fortunately, the 3D U-Net based latent voxel features are trained with coarse occupancy volume losses and naturally fit with voxel-based 3D-GAN losses. Human meshes reconstructed using only the 3D geometry features learned with both coarse occupancy volume losses and 3D-GAN losses are shown in plot-d of Figure 5. Its performances on the DeepHuman benchmark are: CD (2.464), PSD (3.372), Cosine (0.1298), $L2$ (0.4148). Comparing with plot-c, we do observe that "lumps" before the body's belly no longer exist but the mesh exhibits a new problem of having "honeycomb" structures. This phenomenon is also observed in the voxel-based 3D-GAN paper [35]. Moreover, when we fuse such "honeycomb" 3D geometry features with the 2D pixel features to construct aligned query point features, the implicit surface function training fails to converge. We plan to have more studies on this issue as future work.

## 5 Conclusion

We propose to interpolate and fuse geometry and pixel aligned query point features for deep implicit surface function-based single-view clothed human mesh reconstruction. Our method of constructing query point encodings provides dedicated representation for each 3D point that captures its local shape patterns and resolves the feature ambiguity problem. Moreover, the latent voxel features, which we generate by structure-aware 3D U-Nets and supervise with coarse occupancy volume losses, help to regularize the clothed human mesh reconstructions with reduced shape and pose artifacts as well as distortions. The large-scale benchmark ($10\times$ larger then PIFu) and ablation studies demonstrate improved global and local performances of Geo-PIFu than the competing methods. We also discuss typical failure cases and provide some interesting preliminary results on 3D-GAN training to guide future work directions.

## Broader Impact

**Who may benefit from this research?** The VR / AR software developers and 3D graphics designers may benefit from our research. The proposed technique generates single-view clothed human mesh reconstructions with improved global topology regularities and local surface details. Our method can benefit various VR / AR applications that involve reconstructing 3D virtual human avatars for customized user experience, such as conference systems and role-playing games. Moreover, being able to efficiently reconstruct 3D meshes from single-view images is useful for graphics rendering and 3D designs.

**Who may be put at disadvantage from this research?** In the long run, some entry-level graphics artists and designers might be affected. Generally speaking, the 3D gaming and graphics design industries are moving towards automatic content generation techniques. These techniques are not meant to replace highly skilled human workers, but to help improve their productivity at work.

**What are the consequences of failure of the system?** Failed human mesh reconstructions might bring unpleasant user experience. Typical failure cases as well as possible solutions have also been discussed in the main paper.

**Whether the task/method leverages biases in the data?** There might be some biases on human poses and clothes due to long-tail cases. However, our dataset is already $10\times$ larger than the one used in the competing methods. More importantly, our mesh collection procedures can be easily expanded to other domain-specific scenarios to obtain more human meshes with different shapes, poses and clothes to compensate for long-tail cases.

## Acknowledgment

Research supported in part by ONR N00014-17-1-2072, N00014-13-1-034, ARO W911NF-17-1-0304, and Adobe.

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
