[Supplementary Material]

# Supplementary
# Geo-PIFu: Geometry and Pixel Aligned Implicit Functions for Single-view Human Reconstruction

**Tong He**[1], **John Collomosse**[2,3], **Hailin Jin**[2], **Stefano Soatto**[1]

[1]UCLA. [2]Creative Intelligence Lab, Adobe Research. [3]CVSSP, University of Surrey, UK.

{simpleig,soatto}@cs.ucla.edu  {collomos,hljin}@adobe.com

## 1  Implementation Details

Implementation details like network architectures, training/test scripts, pre-trained models, evaluation protocols and *etc.* can be found here *https://github.com/simpleig/Geo-PIFu*. Please check the README.md for step by step instructions.

## 2  More Results

Figure 1: Single-view clothed human mesh reconstruction. Our results have less shape artifacts and distortions than PIFu. PIFu lacks global robustness when hallucinating invisible-side meshes of complex poses and large self-occlusion. Besides improved global regularities and better alignment with ground truth, our meshes also contain more accurate and clear local surface details than PIFu.

| Input | View-1 | View-2 | Input | View-1 | View-2 | Input | View-1 | View-2 |
|-------|--------|--------|-------|--------|--------|-------|--------|--------|

Figure 2: Single-view clothed human mesh reconstruction of our method Geo-PIFu.