[Reviews · NeurIPS 2020]

Review 1

Summary and Contributions: The paper proposed geo-PIFu, an extension or the original PIFu augmented with geometry aligned feature. The geometry aligned features are derived from a 3D U-Net and the these features are concatenated with the sampled PIFu and Z-plane in order to better represent the articulated object. An additional 3D Decoder predicts a coarse volume used as a regularizer during training. The performance is demonstrated on the DeepHuman dataset.

Strengths: 1. The use of latent voxel features as a geometry cue is interesting. 2. The authors show considerable improvement over PIFu. 3. Additional experiments are presented which show the importance of geometry features and early fusion.

Weaknesses: 1. The paper is not compared with SOTA PIFuHD. This comparison is essential as the objectives of the two paper are the same, i.e., improve the reconstruction details. PIFuHD does it via coarse and fine PIFU, whereas the authors here motivate the use of voxel features. It is important to compare the performance of the fine PIFu with geometry PIFu, to truly understand the benefit of geometry. 2. The authors ignore compute considerations. The 3D encoder and decoder will have a large overhead during training, and the 3D encoder will affect runtime performance. In contrary, PIFu does not have 3D convolutions and hence is amenable for real-time applications. The flop count and inference time should be compared to PIFu and PIFuHD. 3. The advantage of having 3D Decoder is not quantitatively described. Overall, the novelty of the aproach is introduction of 3D U-net which is marginal in my opinion as it carries a large computatinal overhead. After having read the rebuttal and review of other reviewers, I maintain my initial assessment. The reasons are two fold: 1. The authors did not provide a fair assessment of the compute cost. I had requested flop count/inference time whereas the authors only reported parameter count. Parameter count is not very meaningful when comparing 2D nets to 3D nets. Without this information, I am skeptical how much of the performance gains stem from architectural novelty vs just adding more compute to the problem. 2. The technical novelty of the paper is further in question in light of the IF-Net paper (not cited) brought into attention by Reviewer#2. The formulation of the principal idea of geometry aligned features is the same as the IF-Net paper. I understand the inputs to the two are different, but the definition, sampling mechanism and implementation share a lot in common.

Correctness: Yes, correct to the best of my understanding. After having read the rebuttal and review of other reviewers, I maintain my initial assessment.

Clarity: The paper is well written, however, there are some typos. A few I could catch are: l75 input than l78 than PIFu

Relation to Prior Work: The method is an extension to PIFu. More recently, PIFuHD has been proposed which is cited but not adequately described. IF-Net by Chibane et.al. is not cited. The main idea of this paper shares a lot in common with the IF-Net paper.

Reproducibility: Yes

Additional Feedback: The comparison to PIFuHD is essential for this work to be useful to the community. The authors of PIFuHD have provided code so it should be straightforward to compare against their method and discuss the insights gained in the process.


Review 2

Summary and Contributions: The authors address the task of reconstructing 3D dressed humans from a single RGB image. The authors present the key insight that directly predicting the detailed 3D surface from a single image is hard due to inherent ambiguities and propose to first predict a coarse 3D volumetric reconstruction and then combine it with image features for final detailed 3D reconstruction using implicit functions.

Strengths: The idea of leveraging coarse 3D shape prediction for single image based reconstruction is technically sound and has been shown to work in previous approaches such as DeepHumans[27], ARCH[7]. It is important to note that these works require additional information; [27] requires coarse 3D shape as input and ARCH requires training a separate model to predict the coarse shape. The proposed method does not require these steps and can be trained directly with raw scans. The paper also reports favourable results as compared to the presented baselines.

Weaknesses: 1. Evaluation setup is not completely clear. Authors do not mention whether they re-trained the baselines (PIFu and DeepHuman) with their images or directly used the pre-trained models. This information is critical as the training data (number of samples and quality of scans) for PIFu was very different than the DeepHuman dataset. 2. It is not clear whether the authors kept the networks comparable for the ablation studies in Table 2. In line 290, authors write that they add several FC layers to the fusion network. It is not clear whether the number of parameters were kept the same for Exp-a,b. One would suspect that multi scale 3D features with more parameters should eventually also capture the features that the 2D branch captures. More details on this experiment would be useful. 3. Authors briefly mention 3D GAN losses in Sec. 4.3 and provide a qualitative example in Fig 5d, but this experiment is not sufficiently explored. It would be useful to add quantitative results in Table 2 (one qualitative example is hardly enough). 4. Authors briefly mention PIFuHD but do not compare against it as a baseline. Though original PIFuHD was trained with high res images, the approach can be meaningfully retrained with regular images as well. This comparison is useful because PIFuHD has significantly better results than PIFu. 5. When works such as ARCH or DeepHuman leverage the coarse 3D reconstruction, they use parametric models such as SMPL (ARCH uses a different model). This greatly constraints the coarse prediction to have a valid human shape (no missing parts, body proportions are respected etc.). In the proposed approach the predicted coarse shape need not have these constraints, hence the predicted coarse shape could still have the problems regarding missing parts (very common when a limb is fully occluded), foreshortening effects and other ambiguities. This makes the argument for using coarse 3D shape weaker. It would be useful to elaborate on this and concretely argue how is making an intermediate unconstrained 3D prediction still useful for the reconstruction (authors do show quantitative results in table 2 but adding some intuitive discussion would be useful, also see pt.2). 6. Sec 3.1.1: The proposed feature learning approach is very similar to IF-Net, Implicit Functions in Feature Space for 3D Shape Reconstruction and Completion, CVPR'20. Chibane et. al. It would be good to cite the paper and highlight the differences in Sec 3.1.1 7. Line 146-147: Authors write that geometry aligned features are good at predicting coarse human shape but not so good for fine details (hence they need the image aligned features) but IF-Net shows that multi-scale grid aligned features are good at making local prediction and key for detailed reconstruction. PiFuHD makes similar argument for their two branch network. It would be useful to elaborate on this further. [MINOR] 1. Line 75: 'then' -> 'than'. 2. Additional references for parametric 3D reconstruction: 2.1 Learning to Reconstruct People in Clothing from a Single RGB Camera, CVPR'19. Alldieck et al. Reconstructs full 3D shape from images. 2.2 Multi-Garment Net: Learning to Dress 3D People from Images, ICCV'19. Bhatnagar et al. Reconstructs body shape + clothing as separate meshes from images. 2.3 3D People: Modeling the Geometry of Dressed Humans, ICCV'19. Pumarola et. al. Use geometry image to reconstruct detailed 3D avatars from images. (2.1, 2.2 can be cited to support the claims in line 106, 107) 3.Line 110: 'on the' -> 'on whether the' 4. Sec 3, In eq. 1 \sigma is used for predicted occupancy, in Eq. 4 it is used for GT occupancy. 5. Line 117, training data should be {I, \sigma(P|I)} not {I, \sigma(P)} as occupancy is a function of both point and the input. Also it is a bit confusing to use \sigma as both occupancy value and a function. Overall, I think this would be a stronger paper if authors address pt. 1,2 and add more discussion regarding the key insights and design choices. ** Post Rebuttal ** Thanks for the rebuttal. + I like the idea of using geometry aligned features and using global shape as a proxy input. + I understand the author's point on not comparing their method against PIFuHD since it was CVPR'20. - My main concern is the technical novelty. The idea of using geometry aligned features (primary contribution of the work) is same as Chibane et al. CVPR'20. Authors address this point in the rebuttal (Chibane et al. use 3D voxels as input where as proposed method works on images) but the formulation is essentially the same. Due to this issue I'm not very confident regarding acceptance.

Correctness: Method is technically sound. It is difficult to access the claims as the evaluation setup is a bit unclear (see Weakness 1,2).

Clarity: I did not have any major issues following the paper. I mentioned some minor points in the Weakness section.

Relation to Prior Work: Prior work is reasonably explored, though comparison with PIFuHD would have been a stronger baseline.

Reproducibility: Yes

Additional Feedback: I did not try it out but the authors submitted code. This is appreciated. I hope the code is made publicly available in case of acceptance.


Review 3

Summary and Contributions: The paper proposes a method for 3d shape reconstruction of humans from a single image. The method builds upon recent works that learn an implicit function representation based on which the 3d geometry is reconstructed using marching cube algorithm. The main contribution lies in designing a geometrically-aligned 3d encoding of the query points based on interpolating latent coarse 3d features. Those are obtained using a 3d U-Net architecture. The authors show that this encoding provides a numerical advantage over previous methods. Moreover, visually, their method is able to reconstruct finer details of the 3d shape.

Strengths: The idea of introducing the 3d embedding of the query points is novel and the authors show a series of ablation experiments to demonstrate its effectiveness. The experimental section includes a detailed discussion on failure cases and comparison with relevant previous works.

Weaknesses: For a fair comparison with previous works, the authors should have also shown results on the publicly available BUFF dataset and (visually) on DeepFashion or on some other natural images. The DeepHuman dataset has indeed more challenging poses and viewing angles but it also seem lower resolution and exhibits some artifacts. Some experimental details are missing: on what dataset is PiFu trained when showing results in table 1? Have the authors re-trained it on DeepHuman dataset? How did they set the network parameters? Rebuttal: - The authors added the clarification on the experimental details. - I am satisfied with their response regarding the comparison with Pifu-HD and Arch methods. - I still believe that their final version should include results on BUFF dataset and DeepFashion, as it would help to better assess their method compared to previous works.

Correctness: The claims and the empirical methodology seem correct.

Clarity: Yes, the paper is generally well written and easy to follow. However, the authors introduce some confusion by referring to the dataset introduced by [27] as DeepHuman, when the name of the dataset should be THuman (as mentioned in [27]).

Relation to Prior Work: Yes.

Reproducibility: No

Additional Feedback:


Review 4

Summary and Contributions: The authors proposed Geo-PiFu, a neural-network based method of reconstructing 3D geometry from monocular RGB images. The method is an extension to the previous work PiFu, addressing its weakness of lacking global structure, with novel contributions to the network design. The authors applied their method to the problem of reconstructing clothed humans, achieving quantitatively and qualitatively better results.

Strengths: 1. Solid contribution to the network design. The authors clearly pointed out the weakness of PiFu, proposed a simple but effective solution, and carried out detailed comparisons with the original method. The proposed method is easy to implement and shares the pixel-aligned idea of PiFu. The proposed method can be seen as a general approach of aggregating local and global features for an implicit-function network, and is compatible with many other following methods based on PiFu. 2. Plausible results. Compared to other methods using same inputs, the results are quantitatively better and visually more accurate especially for the invisible regions, demonstrating the effect of using the proposed global feature.

Weaknesses: 1. Data preparation. As the authors pointed out, data serve a very important role in the whole work. However, the authors did not describe clearly how a) training images are rendered b) query points are sampled during training c) normalizations are applied for 2D and 3D data. Are they the same as PiFu? In implicit function network e.g. PiFu and DeepSDF, both b) and c) are extremely important and could greatly affect the quality of results. 2. Study of global feature. Methods like PiFu purposely avoid using voxel-like feature because of their high computational and memory cost. What is the resolution of the 3D voxel, and does it introduce unnecessary overhead to the whole network? It would be more convincing to study the importance of the global feature in Sec4.2 by comparing with different resolutions of voxel features. Notice when the resolution is reduced to 1x1x1, this is actually the case of using a single global feature.

Correctness: Yes.

Clarity: Generally yes, this paper is well written and easy to understand. The network design is clear and easy to reimplement. However, the authors did not describe in detail for the data preparation, which is reflected in the Weaknesses section.

Relation to Prior Work: Yes, as the point of the paper is mostly around its contributions to the previous work.

Reproducibility: Yes

Additional Feedback: The paper makes a good point of adding the global feature to PiFu, and achieves plausible results. However as stated in section Weakness, the authors might want to include more studies to support their work. The authors promised in the rebuttal they will address these issues and look into a more comprehensive study of their voxel feature representation.

[Author Response · NeurIPS 2020]

• **Shared Questions**
**Are methods reported in Table 1 (main paper) trained with the same DeepHuman dataset?**
Yes, using their GitHub code. We will also release training
/test/evaluation scripts of these competing methods. More-
over, we include results of pre-trained models released by
PIFu and PIFuHD in Table 1. Under the same training data,
PIFuHD achieves lower relative improvement over PIFu
than Geo-PIFu. More discussions on PIFuHD are below.

Table 1: DeepHuman benchmarks. Parameter size of Geo-PIFu is 30616954 (*12 times smaller than PIFuHD*).

| Method | Parameter Size | Mesh | | Normal | |
|--------|---------------|------|------|--------|------|
| | | CD | PSD | Cosine | L2 |
| PIFu | 15604738 | 10.571 | 9.285 | 0.1422 | 0.4141 |
| PIFuHD | 387049625 | 9.489 | 9.349 | 0.1228 | 0.3776 |

**Why not conduct detailed comparisons with PIFuHD in the main paper?**
**1)** PIFuHD was published at CVPR 2020 after the NeurIPS 2020 submission deadline.
**2)** Such a comparison would be unfair. Although not emphasized in their paper, PIFuHD uses ImageNet for pre-training,
additional networks, and higher resolution inputs than all other competing methods and than our method.
**3)** In the limited time available to rebut, we show pre-trained PIFuHD results in Table 1 by upsampling 512x512 images
to 1024x1024 (their code/model). PIFuHD has not released the training code, which involves several stages for multiple
networks. For example: one stack-hourglass for global-PIFu, one stack-hourglass for fine-PIFu, one customized ResNet
for front normal, one customized ResNet for back normal, and one ImageNet-pretrained VGG-16 for perceptual losses.
PIFuHD uses much heavier networks than Geo-PIFu, and requires complex training steps (end-to-end training is even
worse than PIFu). In contrast, we provide all training/test/evaluation scripts to make Geo-PIFu fully reproducible. □
**4)** The two ideas of PIFuHD (using sliding windows to ingest high resolution images, and offline estimated front/back
normal maps to further augment input color images) are both add-on modules *wrt.* (Geo)-PIFu. Given high resolution
images and offline estimated normal maps, one might combine PIFuHD with Geo-PIFu for further improved local
surface details and global topology robustness. But this is out of the scope of our work.
• **Reviewer #1**
**Computation considerations.** Please see Table 1 and answers □, ◯ for more discussions on computation cost of
PIFuHD and Geo-PIFu. Real-time performance is a common challenge for concurrent works, e.g. PIFuHD, ARCH.
**The advantage of 3D Decoder is not quantitatively described.** These results are in *exp-a* of Table 2 (main paper).
• **Reviewer #2**
**More details on the late fusion experiment in Table 2 (main paper).** We add 3 FC layers (112, 224, 256) with Leaky-
ReLU after obtaining the geometry-aligned features for late fusion. These layers introduce additional computation
cost than the early fusion method, which we use in our benchmarks and qualitative demos for its good balance of
computation and performance. More discussions on the capacity of latent voxel features are in answers △ and ◯.
**Quantitative results of the 3D GAN loss.** Mesh: CD (2.464), PSD (3.372). Normal: Cosine (0.1298), $L2$ (0.4148).
We did not include them because they are not the main focus of our work. We will add these results in camera ready.
**Parametric body models.** (Accuracy) Current methods suffer from large pose errors, hurting the rest reconstruction
steps. Thus, DeepHuman has large CD/PSD. (Computation) Although not emphasized in ARCH/DeepHuman papers,
parametric shape estimation networks that they rely on involve many times more computation cost than the rest modules.
**More discussions on IF-Net, CVPR 2020.** While IF-Net takes partial or noisy 3D voxels as input, Geo-PIFu only
utilizes a single-view color image. Thus, IF-Net has access to "free" 3D shape cues of the human subject. But Geo-PIFu
must achieve an ill-posed 2D to 3D learning problem. Meanwhile, Geo-PIFu needs to factorize out pixel domain
nuisances (*e.g.* colors, lighting) in order to robustly recover the underlying dense/continuous occupancy fields. △
• **Reviewer #3**
**More results on other datasets like BUFF and DeepFashion.** The *BUFF test data comprises only 5 front-facing*
*images with simple poses and no self-occlusion*; comparison would not add significant new insight to the existing
evaluation comprising 21744 test images of various poses, camera angles and lighting. We will add visual results on
DeepFashion in camera ready.
• **Reviewer #4**
**Data preparation.** a) Following DeepHuman, we use OpenDR with Lambertian point lights for image rendering. We
will release the rendering scripts. We saved camera pose and lighting settings of each image so that our data can be
reproduced. b) and c) We use the same point sampling strategy and data normalization method as PIFu. Therefore
we can fairly evaluate the impact of our proposed modules. Query points sampling is a critical process that deserves
in-depth studies as a full paper, *e.g.* one emerging work is Curriculum DeepSDF (Duan, Yueqi, et al. ECCV 2020).
**Study of global feature.** The latent voxel feature resolution: (C-8, D-32, H-48, W-32), in total 393216. In comparison,
the latent pixel feature resolution: (C-256, H-128, W-128), in total 4194304. Studying different resolutions of the latent
voxel features is very interesting. We promise to add this experiment in camera ready to make our paper stronger. ◯
**Limited discussions and incremental contribution.** We will explore the directions mentioned in additional feedback
in our future work. As pointed out by the reviewer, many problems are common, open challenges for concurrent works.

[Meta-Review · NeurIPS 2020]

3 reviewers recommend acceptance, 1 is marginally negative. Overall the methodological contributions are valuable but the authors are encouraged to close some of the experimental gaps indicated during reviewing (e.g.compare with PifuHD), better analyse complexity and runtime, and provide a crisper connection to related work (e.g. IFNet paper).